# Spent Grain: A Functional Ingredient for Food Applications

**DOI:** 10.3390/foods12071533

**Published:** 2023-04-04

**Authors:** Ancuța Chetrariu, Adriana Dabija

**Affiliations:** Faculty of Food Engineering, Stefan cel Mare University of Suceava, 720229 Suceava, Romania

**Keywords:** spent grain, valuable by-product, beer industry, whisky, bioactive compound, valorization, circular economy

## Abstract

Spent grain is the solid fraction remaining after wort removal. It is nutritionally rich, composed of fibers—mainly hemicellulose, cellulose, and lignin—proteins, lipids, vitamins, and minerals, and must be managed properly. Spent grain is a by-product with high moisture, high protein and high fiber content and is susceptible to microbial contamination; thus, a suitable, cost-effective, and environmentally friendly valorization method of processing it is required. This by-product is used as a raw material in the production of many other food products—bakery products, pasta, cookies, muffins, wafers, snacks, yogurt or plant-based yogurt alternatives, Frankfurter sausages or fruit beverages—due to its nutritional values. The circular economy is built on waste reduction and the reuse of by-products, which find opportunities in the regeneration and recycling of waste materials and energy that become inputs in other processes and food products. Waste disposal in the food industry has become a major issue in recent years when attempting to maintain hygiene standards and avoid soil, air and water contamination. Fortifying food products with spent grain follows the precepts of the circular bio-economy and industrial symbiosis of strengthening sustainable development. The purpose of this review is to update information on the addition of spent grain to various foods and the influence of spent grain on these foods.

## 1. Introduction

Agro-industrial processes generate substantial amounts of by-products with increased contents of organic compounds that have a significant impact on the environment [1]. On the other hand, the increased demand for food globally drives the discovery of substitute raw materials with affordable pricing and superior nutritional value. By-product valorization has emerged as an important component of food research in recent years [2]. Spent grain (SG) is the main by-product of the beer and distillation industries [3,4]. During whisky and beer production, an average of 8–15 L of effluent and 2.5–3.0 kg of spent grain is generated for each liter of whisky produced, while 0.2 kg of spent grain is generated per liter of beer produced [4,5]. To support the circular economy and the environment, biomass conversion technologies and biorefineries must be developed so that the biomass-based economy grows. Spent grain is a valuable by-product that is rich in nutrients such as dietary fibers (hemicellulose, cellulose and lignin), proteins, monosaccharides (glucose, xylose and arabinosis), minerals, vitamins and lipids [6]. Products enriched with spent grain are called fortified foods [7,8]. Spent grain contains a substantial quantity of bioactive compounds with high antioxidant capacities, including hydroxycinamic acids (particularly ferulic acid and p-cumaric acid) [9,10]. There are several variations of this by-product due to the varied types of cereals, seasons and quality of crops, malting and lautering methods, and composition of the spent grain [11,12,13]. The roles of the functional components of spent grain in the human body are described in Table 1. Due to its high moisture, finding alternative sources of drying spent grain that consume less energy, produces a brighter color and with higher available and digestible protein, is challenging [14]. Spent grain is often used as animal feed due to its nutritional content and low cost; it is used either in wet or dry form [10,15]. Spent grain is also used for the generation of renewable energy to reduce the carbon footprint of alcohol production [16,17].

## 2. Nutritive Value of Spent Grains

Proteins in spent grain can be used as a substitute for fishmeal or soya flour in feed formulations [18]. Approximately 50–60% of dry matter in spent grain is carbohydrates, including glucans, starch, cellulose and arabinoxylans. These carbohydrates can be converted into various biochemical products and biofuels [19,20]. Spent grain can also be used in various biotechnological processes to produce lactic acid, xylitol, microbial enzymes and biopesticides [17,21,22]. The proteins in spent grain contain valuable amino acids, the most abundant of which are glutamine/glutamate and proline [20]. In addition, spent grain is a promising source of lipids, including triglycerides (67% of the total extract), a range of free fatty acids (18%) and lower quantities of monoglycerides (1.6%) and diglycerides (7.7%). Among the fatty acids are linoleic acid (18:2), palmitic acid (16:0) and oleic acid (18:1), as well as small quantities of stearic acid (18:0) and linolenic acid (18:3) [23,24,25]. It is necessary to add value to this by-product and to develop sustainable low-cost methods to make it economically attractive [26]. Harnessing this by-product and developing sustainable processes is an urgent need in food industry [27] (Table 1).

Spent grain is a low cost by-product that is available throughout the year, is nutritionally valuable, rich in fiber, proteins and minerals and can be reused in both food and non-food sectors, including animal feed, compost preparations, biogas production, cultivation of microorganisms and production of biomaterials, biochemicals and bricks [33,37,38]. Spent grain extract is an approved ingredient in food supplements [39]. Spent grain undergoes pre-treatment to make it more accessible. It has a number of advantages, including open cell wall structure, decreased particle size and improved digestibility. Pre-treatments are used in acidic and alkaline environments and in ultrasonic or microwave extractions; however, simple, environmentally friendly, economical and efficient methods are desired. Spent grain is used in food products (Figure 1) for its health promoting effects against constipation, obesity, diabetes and cardiovascular diseases. Phenolics in spent grains are associated with the prevention of chronic cardiovascular and neurogenerative diseases, certain cancers and diabetes. The high fiber content helps in the elimination of cholesterol and fats and improves symptoms of ulcerative colitis [31,33,34,35,36]. Reusing spent grain as a value-added food source for human consumption is appealing because it increases the protein, fiber, vitamin and mineral contents while decreasing the starch and caloric content in grain-based products [40].

Spent grain is a by-product with high nutritional value, thus it is important to identify innovative solutions for returning waste and by-products into the production cycle to obtain innovative quality products [33,41]. A large number of companies have adopted the reduce-reuse-recycle approach, understanding that solving social and environmental issues require changing the strategy of organizations and introducing interdisciplinary actions and methods [38]. The circular economy is based on extending the life cycle of products by reusing, renovating and recycling them for as long as possible, thus reducing waste. In this respect, innovations are stimulated and solutions are found to meet the rising challenges [42,43]. A sustainable approach to the circular economy is necessary in order to use the circular economy structure for food by-products.

Spent grain is a good source of phenolic compounds (ferulic acid, p-coumaric acid, sinapic acid and caffeic acids), which are considered natural antioxidants [44]. There is a growing interest in the valorization of by-products and the circular bioeconomy, by developing alternatives to the conventional use of spent grain for animal feed and compost. The capitalization of food by-products is not only influenced by technical capabilities, but also by socio-economic, supply chain and regulatory factors [45].

The circular and sustainable bioeconomy is gaining traction as a means of addressing climate change and fossilization, increasing resource efficiency and creating new opportunities for long-term economic growth [33].

The aim of this review is to update information on the use of spent grain from the beer and whisky industries in the production of value-added food products. 

## 3. Possible Uses of Spent Grain in Food Products

The food sector is continuously expanding, and consumers are becoming more and more interested in new recipes and healthy diets. The current global context challenges us to find low-cost, high-nutrition, healthy alternatives for food products, and to use industrial by-products. The principal characteristics of several food products enriched with spent grain are presented in Table 2.

### 3.1. Spent Grain in Bread

Bread is one of the most common foods. To obtain bread with acceptable sensorial properties, the amount of spent grain added is limited to 10–15%, as adding a higher quantity leads to a decrease in volume, affects the taste and aroma and changes the rheological properties [46]. Bread enriched with this by-product has a high fiber content, which is associated with health benefits, including increasing digestion and preventing some gastrointestinal diseases. [47]. Not only does the fiber content increase with the increase in spent grain content, the protein content also increases [48]. Furthermore, a high quantity of this by-product influenced the rheological and pasting properties of dough, significantly increased the biaxial extensional viscosity, decreased the strain hardening index as the quantity of substituted flour increased and reduced the uniaxial extensibility. On the other hand, the storage modulus, G″, increased, indicating changes in the structural properties of the dough. These properties negatively affected the baking quality of doughs, conduced to breads with low volumes and dense structures. On the other hand, adding spent grain increased the compositional/nutritional properties, e.g., protein and fiber content [49].

Ktenioudaki et al. [74] produced bread with 15% aded spent grain and sourdough with 15% added spent grain. The samples contained high fiber (11.9% in the spent grain flour added to bread and 12.1% in the spent grain added to the sourdough). As expected, sourdough bread had higher acidity (5.3 compared with 5.8 for breads with no sourdough). The mineral content in the sourdough spent grain samples was higher (107.9 ± 4.8 mg/100 g Ca, 12.7 ± 2.8 mg/100 g Mg, 100.4 ± 17.8 mg/100 g K) compared with bread with no sourdough (98.9 ± 12.5 8 mg/100 g Ca, 11.6 ± 2.0 mg/100 g Mg, 98.9 ± 12.5 mg/100 g K) [74]. Yitayew et al. found that the calcium, magnesium and potassium contents of the bread increased from 76.44 to 150.93 mg/100 g, 87.12 to 176.81 mg/100 g and 116.04 to 225.49 mg/100 g, respectively, as the spent grain quantity increased from 0 to 20% [37]. The in vitro antioxidant activity in the sourdough samples was also higher (132.7 ± 3.0 gallic acid equivalent (mg/100 g sample dwb) for total phenolic content and 83.1 ± 9.3 TEAC (IC50Trolox/IC50Sample) × 10^5^ for DPPH scavenging activity compared with samples with no spent grain sourdough (130.9 ± 4.3 gallic acid equivalent (mg/100 g sample dwb) of total phenolic content, 82.2 ± 9.0 TEAC (IC50Trolox/IC50Sample) × 10^5^ for DPPH scavenging activity) [74]. Similar observations were reported by Aprodu et al. [75] who found higher total phenolic content and antioxidant activity in bread prepared with sourdough. This can be attributed to the action of endogenous enzymes on cell walls. Phytic acid reduces mineral bioavailability through formation of insoluble complexes, but the sourdough fermentation reduces the phytic acid by approximatively 30%, as measured in the bread, hence potentially increasing the bioavailability of minerals in breads containing spent grain [74].

The increase in the concentration of spent grain added to bread decreases its specific volume due to the high amount of arabinoxylans found in spent grain, with an Arabinan:Xylan ratio of about 0.45, a ratio much lower than wheat bran and wheat endosperm (0.88 and 0.67). Sourdough fermentation increased sample volume, but this decreased once spent grain was added [75]. On the other hand, the bread crumb harness increased with addition of spent grain to the wheat flour; however, the bread crumb level was lower in samples with sourdough. This might be attributed to the effect of endoxylanases activity during sourdough fermentation [75]. Adding spent grain increased the starch gelatinization of wheat flour and affected the stability and retrogradation of the starch gels [75]. Shaitan et al. [52] made sourdough bread containing 25%, 50%, 75% and 100% spent grain by fermentation for 8 days. Bread samples prepared with 25% and 50% spent grain sourdough were characterized by higher porosity, acidity and corresponding moisture compared with samples prepared from 100% spent grain sourdough, which had lower porosity and acidity. The spent grain in the samples played a bacteriostatic function; the control sample was the first to show early signs of rope spoilage. The addition of spent grain increased the shelf life of the bread by 24–48 h, thus slowing down the development of rope spoilage in the bread.

Ginindza et al. [76] optimized spent grain in the wheat:maize:spent grain composite flour bread, up to 10%. A higher quantity of spent grain decreased the bread’s specific volume but increased the volume and density. Fiber, protein and ash content increased with the increase in the quantity of spent grain added [76].

Czubaszek et al. [77] found that replacing wheat flour with spent grain reduced gluten yield and deteriorated its quality, leading to a decrease in the sedimentation value and stability, and increasing dough softening. These trends were attributed to the long mixing time and high shear force applied, in addition to the high fiber and protein content [37,50]. Bread with 10% spent grain did not differ significantly from wheat flour bread in terms of appearance, crust, and crumb properties; however, the color of the bread turned from light cream to brown as the spent grain replacement increased [77].

Water absorption ability of the bread increased significantly (from 58.40 to 66.67 mL/100 g) as the quantity of spent grain increased (0–20%), likely due to the increase in high protein and non-starchy polysaccharides in spent grain [37]. Similar results were obtained by Stojceska and Ainsworth [50]. The protein and fiber content of the spent grain also increased the dough development time (from 3.43 to 17.57 min) [37]. Yitayew et al. [37] showed that the loaf weight, volume and specific volume are modified. Increasing spent grain quantity increased the loaf weight (from 127.58 to 148.85 g), while the specific volume of bread loaf decreased (from 2.92 to 2.46 cm^3^/g). The addition of fiber- and protein-rich ingredients increases the hardness of the bread, as fiber and protein absorb a lot of water, leading to a stronger structure. Sensory analysis showed that overall acceptability decreases with increase in spent grain content due to the cumulative effect of the darker color, the taste, the malt aroma and the crumb texture [37].

Sahin et al. [78] used two ingredients derived from spent grain (one rich in fiber and one rich in protein) to make bread. The bread containing additional fibers had high specific volume (3.72–4.66 mL/g), soft crumb texture (4.77–9.03 N) and crumb structure (4.77–9.03 N), whereas bread enriched with protein had increased dough resistance (+150% compared with control sample), which led to a lower specific volume (2.17–4.38 mL/g) and a harder crumb (6.25–36.36 N).

Steinmacher et al. [51] used enzyme-treated and untreated spent grain to produce bread. Enzymatic treatment did not affect the characteristics of the bread, but adding spent grain and enzymes (Pentopan Mono BG and Celluclast BG) directly to the dough improved the texture and volume. Stojceska and Ainsworth [50] used a wider range of enzymes (Maxlife 85, Lipopan Extra, Pentopan Mono BG and Celluclast) to evaluate the characteristics of bread with added spent grain. Increasing the level of fiber in bread has some advantages, including increasing dough development time, dough stability and crumb firmness; however, it also has some disadvantages, including decreased softening and loaf volume. Shelf life, loaf volume and textures improved when Lipopan Extra, Pentopan Mono BG and a mixture of Pentopan Mono and Celluclast were added.

Adding spent grain affected the rheological properties of the dough: the biaxial extensional viscosity was significantly higher in the supplemented doughs. Replacing wheat flour with spent grain significantly reduced the uniaxial extensibility, while the storage modulus (G″) increased, indicating changes in the structural properties of the dough. The strain hardening index decreased as the quantity of substituted spent grain increased [49]. The nutraceutical quality of bread enriched with spent grain is defined by the quantities of antioxidants and fiber. The antioxidant content of the bread increased with the increase in the quantity of spent grain in the bread formulation, as indicated by Baiano et al. [35]. Additionally, the fiber content increased without affecting the structural and sensory attributes of the bread.

### 3.2. Spent Grain in Pasta Products

Pasta is a ready-to-eat product made from durum wheat [79]; however, lately it is also obtained from other flours, flour mixtures with or without additional vegetables or other by-products, resulting in quality products that retain a good consistency after cooking. Several researchers have investigated the partial replacement of flour with ingredients from agro-industrial by-products to make pasta [80,81,82,83,84]. The increasing worldwide consumption of pasta is due to its high digestibility, slow carbohydrate release, relatively low glycemic index compared with bread, pizza or other cereal products, high shelf life, versatility and ease of cooking [56]. Foods that promote health by incorporating ingredients of plant or animal origin during manufacturing are considered nutritional products with added value [85].

In the case of pasta products, the addition of spent grain does not have a major influence on the functional properties, even at concentrations of 25%, which makes adding spent grain to these products in order to increase their nutrient content acceptable [46]. Nocente et al. conducted a study on pasta formulations by adding spent grain from two species of cereals (einkorn and tritordeum), resulting in pasta with noticeably higher protein, fiber and β-glucan content and, to a minor extent, increased antioxidant capacity and good sensory quality [53]. In another study, Nocente et al. [86] used a blend of semolina and spent grain to produce spent grain-enriched pasta characterized by high fiber and antioxidant content. Schettino et al. used bioprocessed spent grain to obtain fortified pasta labeled “High fiber” and “Source of protein” [55]. Cuomo et al. [54] used two fractions of spent grain (5–10% protein and 10–20% fiber) to obtain high fiber and high protein pasta. Several features of pasta were evaluated, including proximate composition, color, optimal cooking time, sensory features and texture. Protein-enriched pasta had a protein content of about 18% and a fiber content greater than 8%, meaning that it contained about 30% of the protein content recommended by EFSA [87]. Food products can be classified into “fiber source” products, which contain at least 3 g dietary fiber/100 g product, and “fiber-rich” products, which contain at least 6 g dietary fiber/100 g product according to Regulation (EC) No. 1924/2006 [80]. As expected, the protein- and fiber-enriched pasta had a darker color. The L* (brightness) parameter showed a significant reduction compared with the wholegrain semolina paste sample [46,54,86,88], as shown in Figure 2.

Optimal cooking time was between 11 min and 13 min 30 s. Values close to the blank samples and firmness of pasta were positively evaluated. The firmness characteristics, evaluated using instrumental analysis, were associated with sensorial features and were very close to each other [54]. Cappa and Alamprese [80] added egg white powder to spent grain pasta to improve the structural properties of fresh-egg pasta (lasagna) but the mechanical properties were poor.

Sahin et al. [15] developed enriched pasta containing protein and fiber fractions derived from spent grain, resulting products with stronger gluten networks and bonding properties, compact structure, higher firmness and higher tensile strength, but lower glycemic index. An important criterion for pasta quality is cooking loss, with low cooking loss (CL) being more desirable. Pasta labeled high in fiber had lower CL values (3.47 ± 0.86%). Increasing the quantity of ingredients derived from spent grain decreased cooking loss [15]. Starch gelatinization and protein coagulation are two processes responsible for the formation of the structure and quality of pasta during cooking. Adding ingredients derived from spent grain decreased the degree of starch gelatinization due to the high protein content [89]. Starch is physically captured in a protein matrix due to its interaction with proteins through molecular forces (ionic, hydrogen and covalent bonds), which leads to a reduction in the degree of gelatinization and an increase in resistance to shearing and heat. The increased quantities of starch and protein compete for water, leading to reduced swelling of the starch during gelatinization. Reducing the amount of total starch affects the degree of gelatinization and the cooking loss of pasta [15].

Nocente et al. [86] showed that the total antioxidant capacity in spent grain-enriched pasta increases as the quantity of spent grain increases. Most antioxidant compounds (phenolic acids and other polyphenols) are found in the outer layers of the barley grain and in the aleuronic layer of the kernels.

Optimal cooking time reduced after pasta fortification, probably due to the increase in dietary fiber content, which alters the structure of pasta and permits early starch gelatinization and accelerates water penetration [90]. These findings were in agreement with the findings by Nocente et al. [86]. Other quality parameters of the pasta include the amount of water absorbed (WA) by the pasta during the optimal cooking time—associated with the swelling and gelatinization of the starch. Good quality pasta has WA of 150–200 g water/100 g pasta. The swelling index (SI) gives us information about the integrity of the protein matrix, which restricts water penetration [89,91]. The increase in the swelling index is due to a weakened gluten network, which allows increased amounts of water to enter the starch granules, leading to faster gelatinization. Good quality pasta has SI values of approximately 1.8 [90]. These data are in agreement with those of the study by Chetrariu and Dabija [56].

Total organic matter increases with the increase in the spent grain content added to the recipe, possibly due to the high fiber content and the weakened gluten network, leading to swelling of the starch granules and release of a higher quantity of starch while cooking the pasta [86]. Very good quality pasta has total organic matter values lower than 1.4, good quality pasta has values between 1.4 and 2.1, and values higher than 2.1 represent poor quality pasta [91]. Microscopic analysis of the pasta showed a continuous protein network, with protein aggregates of different sizes derived from the spent grain. The degree of starch swelling also increased in the outer part of the sample but decreased towards the core [55].

Good quality pasta must have several attributes: moderate optimum cooking time, low cooking loss, water absorption and swelling index, moderate increase in volume with firmness and high chewiness and low adhesiveness, given by a consolidated and non-continuous protein matrix. This limits the swelling of starch and makes the diffusion of water to the core of the pasta difficult, leading to greater retention of amylose in the structure and less amylopectin on the surface [91]. All these studies show that spent grain can be used in a saturated market to produce innovative products; the ingredients used represent a stable and sustainable solution for spent grain upcycling.

### 3.3. Cookies and Shortbreads

Cookies are food products to which different flours can be added because they accommodate a wide variety of formulations and ingredients, are ready-to-eat, represent a good source of energy, have a long shelf life and are accepted by consumers of all ages [92]. Adding 20% spent grain to the cookie formulation increased the protein content by 55%, the lysine content by 90% and the fiber content by 220% compared with the control sample [93]. The addition of spent grain to cookies is proportional to the increase in dietary fiber content [13] and also depends on the particle size of the spent grain. Öztürk et al. [57] studied the influence of spent grain particle size on the quality of cookies. Medium and coarse particle sizes resulted in better properties in terms of spread ratio, hunter color values and overall sensory scores compared with cookies made with spent grain of fine particle size. Dough development time and dough stability increased with spent grain substitution level, resulting in higher energy costs. Adding spent grain increased the phenolic acid concentration and ferulic acid was predominant in all cookies. A 20% quantity of spent grain resulted in lower hydrolysis and glycemic index, and less total starch content compared with the control cookies [58]. Fat is needed for cookie production; as the quantity of substituted spent grain increased, fat levels in the cookies also increase. Additionally, the thickness and width of cookies increased with the addition of spent grain, while the spread ratio decreased in comparison with control samples [59]. Figure 3 shows cookies produced using 10% spent grain. Given that the replacement of wheat flour with spent grain in the cookie recipe increases the proportional fiber and protein content, the use of spent grain becomes promising for groups of consumers with nutritional deficiencies [60].

Petrović et al. [61] evaluated the effect of fresh spent grain (milled and non-milled) in cookie formulations on cookie quality parameters. Fresh spent grain had no negative effects on microbiological stability, and adding 25% of it produced the best sensory characteristics (appearance, hardness, grittiness and flavor). Fresh spent grain was susceptible to microbial attack and chemical damage due to its chemical composition, but the study shows that a proliferation of microbiological compounds (Yeast and molds, *Escherichia coli* and *Clostridium* spp.) did not occur. The high quantity of fiber and protein in spent grain increased water absorption, negatively impacting the hardness and chewiness.

Replacing 30% of wheat flour with spent grain flour in the shortbread recipe led to a significant increase in the fiber (particularly arabinoxylans) and protein content, and a decrease in carbohydrate levels and energy compared with the control samples. The replacement also showed acceptable sensory characteristics [29].

### 3.4. Muffins

Shih et al. looked into how two drying techniques (impingement and hot-air drying) affect the composition of spent grain and the quality of muffins made using spent grain. The study showed that impingement-dried spent grain may be used as a functional component in muffins to enhance the value to the food chain and to provide nutritional and environmental benefits. The study also found that adding spent grain flours to muffins (substituting 15% of the wheat flour) increased their protein and total dietary fiber contents by 23% and 13%, respectively, without influencing consumer acceptance of the products. In general, due to the higher concentrations of these nutrients in spent grain flour compared with wheat flour, the amount of fat, protein and total dietary fiber in fortified products is significantly higher [62]. The viscosity of the batter increases as the amount of spent grain in the muffin recipe increases. This may be because the spent grain has a high fiber content that acts as a thickening agent by absorbing water in the batter. A study of 18 participants who consumed muffins with 30% added spent grain daily for 8 weeks showed beneficial effects associated with reduced systolic blood pressure and insulin compared with the control group [63]. Another study on the use of spent grain in muffins found that 30% spent grain retains consumer approval and offers more chances of triggering biological reactions due to the higher levels of proteins, fiber and antioxidants. Furthermore, nutrient content could be mentioned on the labels of muffins with 20% and 30% spent grain, both of which are considered “good sources” of protein and fiber, because the portion size contains more than 10% of the recommended daily value of each nutrient [40].

### 3.5. Wafers

Wafers come in a wide variety of assortments and are obtained by baking special forms of fluid dough consisting of wheat flour, water, salt, aeration agents and other ingredients used to add taste and aroma, and are presented in the form of sheets or different alveoli formats [94] with high porosity and no filling (Figure 4). The disadvantages lie in the fact that parts of these products may have low nutritional values and poorly defined sensory characteristics.

Gumminess, chewiness, springiness, firmness and cohesiveness increased in the spent grain sample, while adhesiveness decreased with the addition of spent grain [64]. Water activity is an important instrumental measure of crispiness in wafers and should be between 0.387 and 0.52 [65]. Two regions of wafers with different porosities can be highlighted: a dense part called “skin” and a less porous part called “core”, which is the central part of the baked sheet [66]. Analyzing the microstructure of the wafers is important for determining the quality of the products and involves measuring the size of the pores and cell wall sizes throughout the cross-section. This analysis shows that the distribution of the pores on the cross section is heterogeneous, with the center of the wafers having larger pores and the edges having smaller pores and denser skins [66].

### 3.6. Snacks

Nagy and Diósi obtained products with added spent grain that had positive nutritional values, including increased total content of polyphenols, flavonoids, proteins, fats, dietary fiber and energy, compared with the control samples [48]. Ktenioudaki et al. [95] obtained crispy snack slices containing 10% spent grain with a high crispiness index and low crispiness, indicating that this quantity of added spent grain did not negatively affect the crispiness of the finished product. Adding a higher quantity of spent grain modified the texture and crumb structure and altered the odor profile. Crispy slices containing 10% spent grain were highly acceptable to panelists compared with the control samples. This quantity of added spent grain almost doubled the fiber content of the baked snacks [95]. Crofton and Scanell conducted a study of four types of spent grain snacks; the crispy cracker snack was the most preferred, followed by the crispy sticks with dip, the fruity biscuits and finally the twisted breadsticks [96]. Stojceska et al. [67] studied the effects of spent grain on the textural and functional properties of extrudates and found that at between 10% and 30% reduced cell size, the expansion of the product reduced and the phytic acid content and bulk density increased. The optimal level of spent grain was set at 20% in order to obtain products similar to those available on the market, although a substitution of 30% still led to products with acceptable physico-chemical characteristics. The textural properties were adjusted by incorporating starch and specific mechanical energy and controlling extrusion parameters conduced to an acceptable expanded ready-to-eat snack. Ainsworth et al. [68] conducted a similar study on the effect of brewers spent grain and screw speed on the selected physical and nutritional properties of an extruded snack. They found that phytic acid and the resistant starch content of the samples increased significantly with the addition of up to 30% of spent grain to the formulation, although screw speed had no significant effect on total antioxidant capacity and total phenolic compounds. Kirjoranta et al. [97] conducted a study on the effects of spent grain on process parameters of snacks and concluded that 10% of spent grain increased hardness and caused a small expansion. The expansion increased with increasing screw speed and decreasing water content. Several ways of achieving a greater expansion of extruded snack include adding starch, enzymatic treatment of spent grain to solubilize part of the insoluble dietary fiber or grinding spent grain into small particles. The snack market is rapidly expanding with the frequent introduction of innovative bars fortified with proteins, fibers and other rich nutrients such as spent grain [98].

### 3.7. Extruded Spent Grain

Extrusion is a relatively new method of cooking that involves continuous mixing, cooking and extrusion of food products. The extrusion process takes place in an extruder in which thermo-mechanical processing takes place. The temperature inside the extruder is increased by subjecting the material to high compressive and shearing forces. This process leads to cooking of the product and has the advantage of immediate and efficient modification, giving a product with superior quality [66]. Extrusion is a thermo mechanical process that combines several unitary operations representing a viable, transferable opportunity with a beneficial impact on the functional, technological and food safety characteristics of the product [99]. The extrusion process can improve the balance between soluble and insoluble dietary fiber contents, breaking polysaccharide bonds under mechanical stress, and releasing the content of phenolic compounds trapped in the dietary fibers through shearing. Among the undesirable aspects of extrusion is the Maillard reaction, which favors the production of acrylamide or the reduction in the content of essential amino acids; aspects that depend on the required processing conditions [100]. Gutiérrez-Barrutia et al. studied the effect of extrusion on spent grain and found that extrusion had positive effects on spent grain, increasing the content of soluble dietary fibers, changes caused by the thermo mechanical process that can disrupt the cell wall matrix resulting in smaller and more soluble fragments. Extruded spent grain can be considered a suitable ingredient for human consumption from a microbiological point of view [100].

### 3.8. Yogurt and Plant-Based Yogurt Alternatives

Various quantities of spent grain were used as substitutes for yogurt fermentation, and the effects on microstructural characteristics such as surface chemical characteristics and confocal microstructures were investigated. Yogurt’s syneresis level was considerably reduced when spent grain was added. Adding spent grain decreased fermentation time and increased viscosity and shear stress. As a result of amino acids being released, the proteolytic action of the additional microorganisms in yogurt manufacture shortens the fermentation process by enhancing microbial growth. The reduced fermentation period was also influenced by the inclusion of protein and fat substitutes. Yogurt’s maximum quality, including its acidity, rheological behavior and lactic acid bacteria development improved with the increase in the added spent grain from 5% to 10%. While 15–20% of spent grain gave the lowest syneresis while producing the same amount of acidity and lactic acid bacteria, it reduced the yogurt’s flow performance [69].

Spent grain has a great water-holding capacity due to its significant insoluble dietary fiber content (particularly arabinoxylans, which have a high capacity to bind water and play a potential prebiotic role). Due to these properties, spent grain may be able to control the behavior of semi-solid foods and hence replace the need for starch. In the study conducted by Naibaho et al., spent grain flour and three different protein extracts from spent grain added to plant-based yogurt-alternatives maintained textural and gelling formation, while increasing shear stress and viscosity [70].

### 3.9. Other Food Products

Spent grain can also be used as breadcrumbs for schnitzel, in fillings for vegetable burgers and in Frankfurter sausages [47]. Nine experimental Frankfurters were made using spent grain of three distinct particle sizes—fine (212 µm), medium (212–425 µm), and coarse (425–850 µm). As expected, the Frankfurters’ total dietary fiber content improved as more spent grain was added. The water-holding capacity was correlated with the total dietary fiber in the experimental samples, and the total dietary fiber content of the samples generated with coarse-particle-size (425–850 µm) spent grain were higher than those of the other samples. The amount of spent grain and the low level of fat in the coarse, medium and small particle size groups appeared to have a negative impact on all textural criteria, with the exception of springiness. The study revealed that spent grain has great potential as a source of dietary fiber and might be used as a fat substitute to create meat products that are both high in dietary fiber and low in fat. Most of the textural and sensory criteria were substantially correlated with overall acceptance, based on statistical cluster analysis [71].

Özboy-Özbaş et al. showed that spent grain can be used as an ingredient in tarhana, a fermented wheat flour-yoghurt product, with acceptable results obtained using 6% spent grain. This quantity of spent grain increases the protein and fiber content while maintaining sensory qualities within acceptable limits [72].

McCarthy et al. used different methods to introduce phenolic extracts into fruit beverages (fruit juice and smoothies). The maximum concentration of phenolic extract in spent grain was 10%, considerably raising the FRAP activity of cranberry juice and demonstrating the possibility of using spent grain phenolic extracts as antioxidants in functional foods [73].

Spent grain has replaced flour/semolina in the traditional Herzegovinian product, Cupter. This product is made from wheat flour or semolina and grape must and is described as a sweet jelly. Spent grain influences the odor, color, texture and flavor profile of the traditional product, the taste of which is well known to consumers [101].

Spent grain can also be used to make edible coating composite films for fresh strawberries by immersing the strawberries in coating solution for 2 min before testing weight loss, pH, dry matter and anthocyanin content over 5 days. The carboxyl methylcellulose edible composite film positively affected the appearance of strawberries after the testing period, preserving their freshness for a longer period compared with uncoated strawberries. The strawberries had low weight loss because the films prevent moisture loss. No significant differences in anthocyanin levels were observed between coated and uncoated strawberries. The pH levels of the coated and uncoated strawberry samples also showed little variation over the course of storage [102].

## 4. Conclusions

The use of spent grain as an ingredient in finished food products is an opportunity to reduce by-products in the beer and whisky industries while improving the nutritional content of the food obtained. Studies showed that consumer acceptability limits for food products comparable to commercial ones fall within a 20% concentration of the spent grain, although 10–15% spent grain is considered optimal for sensory properties. By-products of the agro-industrial sector are important resources that can be used as raw materials to create food products with added value, supporting the circular economy. One of the basic tenets of the circular economy and one of the biggest problems in food engineering in recent years has been the sustainable use of organic waste and agri-food by-products. Due to its high quantity and low cost, spent grain is a source worth exploiting. The growing demand for products with stable ingredients obtained from food by-products is stimulating the identification of innovative alternatives. Additionally, food industry by-products are a good source of proteins, minerals, fatty acids, fiber and bioactive substances that can prevent nutrition-related disorders and improve consumers’ physical and mental well-being. The global demand for food is rising, driving researchers to look for alternative raw materials with good nutritional value. There is a pressing need to utilize this by-product and to create sustainable methods of using it. The reuse of spent grain in food products (muffins, cakes, biscuits, etc.) brings both economic and environmental benefits, thus reducing pollution. In subsequent studies, we aim to determine the upper limit of spent grain that can be incorporated into food products, as well as to identify their advantages and disadvantages.

## Figures and Tables

**Figure 1 foods-12-01533-f001:**
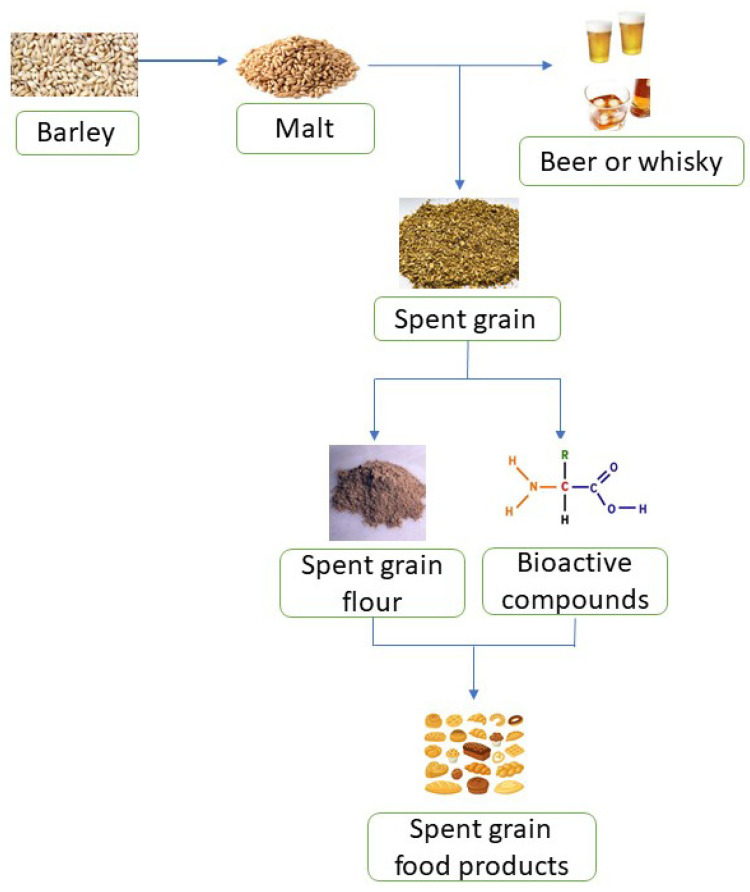
Synthetic process of generating and using spent grain to produce food products.

**Figure 2 foods-12-01533-f002:**
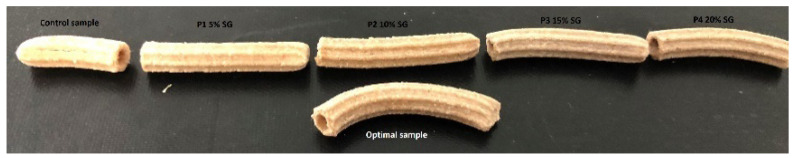
Spent grain pasta [84].

**Figure 3 foods-12-01533-f003:**
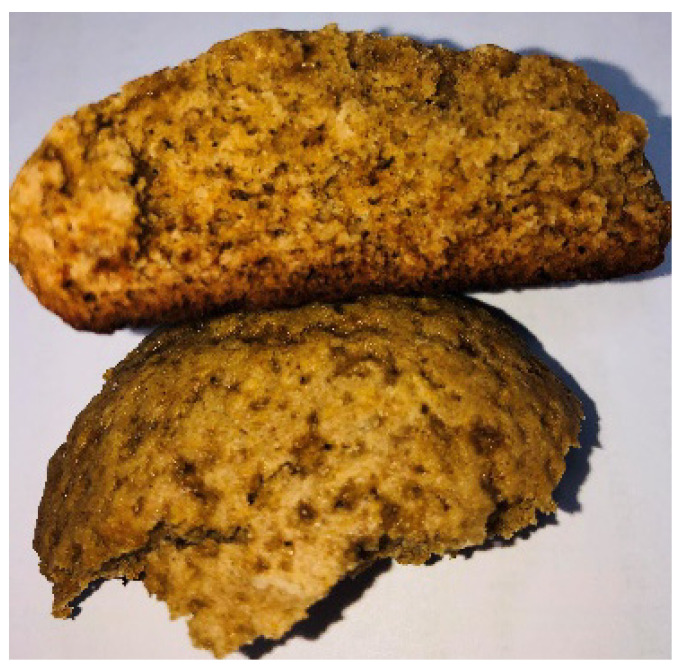
Spent grain cookies.

**Figure 4 foods-12-01533-f004:**
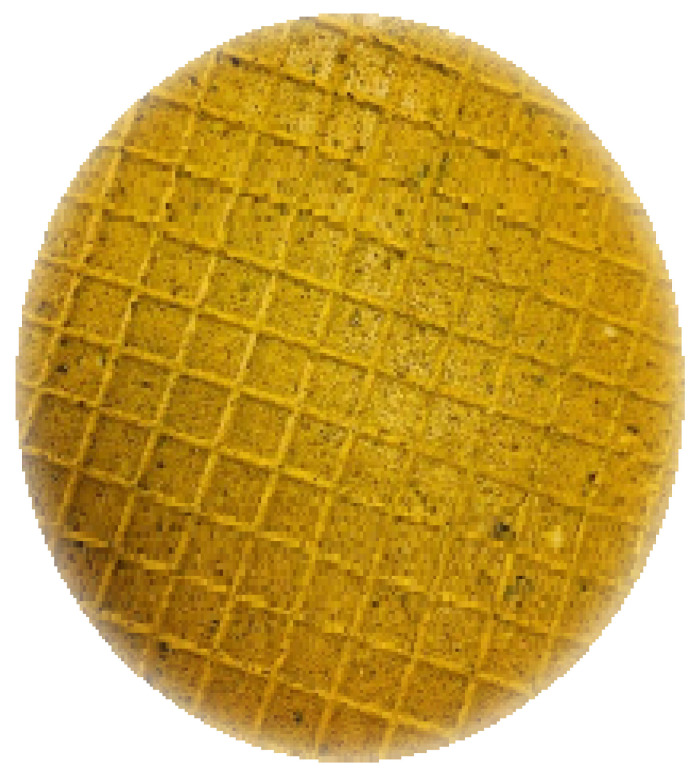
Spent grain wafers.

**Table 1 foods-12-01533-t001:** Functional compounds in spent grain.

Functional Compounds in Spent Grain	Role in Human Body	Study
Non-cellulosic polysaccharides (β-glucan and arabinoxylans)	β-glucan reduce the rise in blood glucose after meals; Arabinoxylans reduce blood glucose levels.	[28,29]
Proteins	Increase satiety; Regulate long-term energy balance.	[30,31]
Polyphenols	Anti-carcinogenic; Anti-inflammatory and antioxidant activities.	[31,32,33,34]
Fiber	Dietary fiber reduces cholesterol levels; Increases fecal bulk.	[28,34,35,36]
Vitamins	Vitamins have antioxidant properties.	[28]

**Table 2 foods-12-01533-t002:** Characteristics of functional foods derived from spent grain.

Functional Foods derived from Spent Grain	Quantity of Spent Grain Added	Properties	Study
Bread	10–15% spent grain flour	Acceptable sensorial properties; High fiber content (health benefit); Increased mineral content; Influences the rheological and pasting properties of dough; The biaxial extensional viscosity is significantly higher; The strain-hardening index decreases with increasing quantities of flour substitution; Reduces the uniaxial extensibility, while the storage modulus, G″, increases; Addition of spent grain increases the composition/nutritional properties; The color of bread turned from light cream to brown; Water absorption increases with the quantity of spent grain; Increased crumb firmness; Increased antioxidant content.	[35,37,46,47,48,49,50,51]
Bread obtained from fermented spent grain	25%, 50%, 75%, 100% spent grain sourdough	Changes the porosity and acidity; Bacteriostatic function (the shelf life of bread increases).	[52]
Spent grain pasta	5–25% spent grain flour	Increased protein, fiber and β-glucan content; Increased antioxidant content; The higher the spent grain content, the darker the color of the pasta; A compact structure with higher firmness; Decreased cooking loss; Decreased degree of starch gelatinization; Reduced the optimal cooking time; Increased total organic matter.	[53,54,55,56]
Cookies	Max 30% spent grain added	Fiber and protein content increases; Dough development time and dough stability increases; Total antioxidant activity increases; Water absorption increases.	[57,58,59,60,61]
Shortbread	30%	Increase in fiber and protein content; Decrease in carbohydrate levels and energy value.	[29]
Muffins	15–30%	Increases the amount of fat, protein and total dietary fiber; Increases the viscosity of the batter.	[62,63]
Wafers	5–20%	Gumminess, chewiness, springiness, firmness and cohesiveness increase.	[64,65,66]
Snacks	10–30%	Increase the total content of polyphenols, flavonoids, proteins, fats, dietary fiber and energy; Increases phytic acid and resistant starch content.	[48,67,68]
Yogurt and plant-based yogurt alternetives	5–20%	Yogurt’s syneresis level was considerably reduced; Decreased fermentation time and increased viscosity and shear stress; Maintained textural and gelling formation.	[69,70]
Frankfurters sausages	1–5%	Total dietary fiber increases.	[71]
Tarhana	6%	Increase in protein and fiber content.	[72]
Fruit juice and smoothies	0–10%	Increased antioxidant activity.	[73]

## Data Availability

No new data were created or analyzed in this study. Data sharing is not applicable to this article.

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
