# Peer review of "Spent Grain: A Functional Ingredient for Food Applications"

_foods, 2023, doi:10.3390/foods12071533_

Round 1

Reviewer 1 Report

The present review manuscript covers the application of spent grains as food application. The efficient utilization of these spent grains from the brewing industry could significantly impact the food industry and improve the economics of the brewing industry. The language is clear and easy to understand. The hypothesis is sound and clear.

My observations are as follows-

       i.          Abstract: please mention some important uses/ potential uses of spent grain

     ii.          Keyword: have scope for improvement as spent grain, bear industry, ----

   iii.          L 30-31: Data or estimations or projections (if available or calculated) of the total amount of spent grains produced per year and by proper utilization how much we can save the environment and economics will strengthen the hypothesis.

   iv.          L41: in order to consume less energy? Please explain as the dry matter will have more energy.

     v.          Introduction part could be split into the introduction and nutritive value of spent grains; so as to minimize some repetitions mentioning the nutritive quality of spent grains.

   vi.          L88: please define the circular economy

  vii.          L125: SG means spent grain?

viii.          L410 Reference number

   ix.          I would recommend a tabular presentation of some important food products formulated with spent grain along with the optimum level, salinet faindings etc.

Author Response

Response to Reviewer 1

Dear Referee,  

We would like to thank the referee for the close reading and for the proper suggestions.

We hope that we provide all the answers to the reviewer’s comments.

The present review manuscript covers the application of spent grains as food application. The efficient utilization of these spent grains from the brewing industry could significantly impact the food industry and improve the economics of the brewing industry. The language is clear and easy to understand. The hypothesis is sound and clear.

First of all, we would like to thank the referee for the close reading and for all the given comments suitable for improving the manuscript.

My observations are as follows-

 Point 1: Abstract: please mention some important uses/ potential uses of spent grain

We would like to thank to the referee for her/his suggestions in order to improve the quality of the manuscript.

Point 2: Keyword: have scope for improvement as spent grain, bear industry, ----

Thank you, we added some keywords.

Point 3: L 30-31: Data or estimations or projections (if available or calculated) of the total amount of spent grains produced per year and by proper utilization how much we can save the environment and economics will strengthen the hypothesis.

We want to thank to referee for her/his suggestion.

Point 4: L41: in order to consume less energy? Please explain as the dry matter will have more energy.

In this paragraph we refer to the fact that spent grain has an increased moisture content (about 80%), and in order to reduce its moisture and make it a microbiologically stable product, it needs to be subjected to the drying process (in a conventional oven using electricity or alternative sources).

Point 5: Introduction part could be split into the introduction and nutritive value of spent grains; so as to minimize some repetitions mentioning the nutritive quality of spent grains.

We want to thank to referee for her/his suggestion. We split the Introduction part, it looks better now.

Point 6: L88: please define the circular economy

We defined circular economy as „based on the extension of the life cycle of the products by reusing, renovating and recycling them for as long as possible, thus reducing the waste to a minimum. In this respect, the innovative part is stimulated and solutions are found in front of the challenges that have arisen.”

Point 7: L125: SG means spent grain?

Yes, SG mean spent grain.

Point 8: L410 Reference number

Thank you, we added the reference number.

Point 9: I would recommend a tabular presentation of some important food products formulated with spent grain along with the optimum level, salinet faindings etc.

Thank you for your remark, we added a Table in the manuscript.

Reviewer 2 Report

The submitted review is about possible ways of re-using cereal-based by-products for formulation of enriched staple foods. It need be revised. The main reason is due to the too simplistic way the authors addressed this topic. For example, very little is described about required treatment of by-products before being correctly included in enriched foods: e.g., removal of compounds and/or microorganisms from previous processing, enzymatic treatment, etc. Also, how can spent grains affect the rheological properties of enriched food? Authors described the use of by-products in pasta without addressing the problem of drying. 

It looks as a mere list of foods where spent grains can be included without facing drawbacks and limits. Nutritional added values cannot be the only driving force for this.

Author Response

Response to Reviewer 2

Dear Referee,  

We would like to thank the referee for the close reading and for the proper suggestions.

We hope that we provide all the answers to the reviewer’s comments.

From my point of view, the publication is interesting and enriches the knowledge in the field of "green technologies".

The only complaint I can make is the lack of clarity as to exactly how the BSG is used in the production of yogurt. Please provide a brief description of the technological process for this bran application.

First of all, we would like to thank the referee for the close reading of the manuscript. The solid part resulting from the filtering is called spent grain, which is subjected to the drying process up to a moisture of 2-5%, grounded in a laboratory mill and sieved through Ë‚250 µm sieving. Then the milk was mixed with spent grain (0:100, 5:95, 10:90, 15:85, 20:80, spent grain:milk, w/w) and mixed properly. The mixture was pasteurized, microbial cultures were added, and subsequently thermostated to pH between 4.3 and 4.8.

Reviewer 3 Report

From my point of view, the publication is interesting and enriches the knowledge in the field of "green technologies".
The only complaint I can make is the lack of clarity as to exactly how the BSG is used in the production of yogurt. Please provide a brief description of the technological process for this bran application.

Author Response

Response to Reviewer 3

Dear Referee,  

We would like to thank the referee for the close reading and for the proper suggestions.

We hope that we provide all the answers to the reviewer’s comments.

From my point of view, the publication is interesting and enriches the knowledge in the field of "green technologies".

The only complaint I can make is the lack of clarity as to exactly how the BSG is used in the production of yogurt. Please provide a brief description of the technological process for this bran application.

First of all, we would like to thank the referee for the close reading of the manuscript. The solid part resulting from the filtering is called spent grain, which is subjected to the drying process up to a moisture of 2-5%, grounded in a laboratory mill and sieved through Ë‚250 µm sieving. Then the milk was mixed with spent grain (0:100, 5:95, 10:90, 15:85, 20:80, spent grain:milk, w/w) and mixed properly. The mixture was pasteurized, microbial cultures were added, and subsequently thermostated to pH between 4.3 and 4.8.

Reviewer 4 Report

Manuscript ID: foods-2287996

Title: Spent Grain - a Functional Ingredient for Food Applications

Dear Editor

The manuscript is about the functional foods containing spent grain and well written. However, needs minor revision to improve its quality according to the comments:

1.     Please insert a Table and summarize the functional foods containing spent grain and its effect on their main properties.

2.     Please use either the abbreviation or word in all manuscript. For instance, use SG instead of spent grain.

3.     I suggest the authors to discuss about the pre-treatments to improve the spent grain functionality and attributes.

4.     Conclusion should be rewritten, and highlight authors viewpoints and futures perspective.

5.     Please use subsections and heading for each functional food.

Author Response

Response to Reviewer 3

Dear Referee,  

We would like to thank the referee for the close reading and for the proper suggestions.

We hope that we provide all the answers to the reviewer’s comments.

The manuscript is about the functional foods containing spent grain and well written. However, needs minor revision to improve its quality according to the comments:

  1. Please insert a Table and summarize the functional foods containing spent grain and its effect on their main properties.

We would like to thank to the referee for her/his remarks and for careful reading of our manuscript. We added a Table in the manuscript.

  1. Please use either the abbreviation or word in all manuscript. For instance, use SG instead of spent grain.

Thank you for your suggestion. We made the changes.

  1. I suggest the authors to discuss about the pre-treatments to improve the spent grain functionality and attributes.

We added some informations about pre-treatments.

  1. Conclusion should be rewritten, and highlight authors viewpoints and futures perspective.

 We would like to thank to the referee for her/his suggestions in order to improve the quality of the manuscript. We modified the Conclusion section.

  1. Please use subsections and heading for each functional food.

We would like to thank to the referee for her/his remarks. We made the changes.

Round 2

Reviewer 2 Report

I do not have further comments or request for the revised version of the Manuscript. I checked all changes in the manuscript and I confirm that It can be accepted in this present form.